# Parameter Estimation for Uniformly Accelerating Moving Target in the Forward Scatter Radar Network

Xiaofeng Ai, Yuqing Zheng *, Zhiming Xu and Feng Zhao

State Key Laboratory of Complex Electromagnetic Environment Effects on Electronics and Information System, National University of Defense Technology, Changsha 564211, China; aixiaofeng@nudt.edu.cn (X.A.); zhimingxu@nudt.edu.cn (Z.X.); zhaofeng321@nudt.edu.cn (F.Z.)

\* Correspondence: zhengyuqing@nudt.edu.cn; Tel.: +15274980424

**Abstract:** Passive radar based on the global navigation satellite positioning system (GNSS) has become the focus of attention in the field of radar. A parameter estimation method is proposed in the forward scatter radar (FSR) network based on GNSS to extend the application scenarios. For uniformly accelerating moving targets, only the instant times when the target crosses the individual baselines are used to retrieve the target motion parameters. The target position, velocity, and acceleration information can be obtained. Firstly, the minimum network configuration is derived theoretically. Then, the effects of crossing time error, station location error, transmitting/receiving station deployment, and target height on the accuracy are analyzed through Monte Carlo simulations. Finally, the simulation results indicate that the target position estimation error is in the order of 100 m. This paper provides the fundamental theory of aerial target positioning with a GNSS-based FSR network.

**Keywords:** global navigation satellite positioning system; forward scatter radar; parameter estimation; crossing time; passive radar

## 1. Introduction

Passive radar based on GNSS has been researched for more than twenty years. GNSS-based passive radar has the advantages of wide coverage [1,2] and good concealment. However, the great challenge is that the scattering signal is extremely weak. Long-term accumulation and super large aperture antenna are needed for long-distance target detection. This leads to high complexity and huge cost.

As for the detection capability of the GNSS-based passive radar system in practical scenarios, some experimental verification results of various countries are summarized in Table 1, from when global position system (GPS) signals were used to detect civil aircraft, military aircraft, and Russian space stations in 1995 [3]. With the support of the European navigation system authority, a project named "*Galileo-based passive radar system for maritime surveillance-spyglass*" was established in 2015. In the actual experiments, two different maritime targets were detected by up to 12 GNSS satellites belonging to different satellite constellations (GPS, Global Navigation Satellite System (GLONASS), and Galileo) simultaneously. Among these satellite constellations, the time difference of arrival (TDOA) location method is used to obtain the target position. The long-term moving target indicator (MTI) processing of merchant ships is realized, and the detection distance can achieve 3 km [4–6]. Table 1 indicates that the detection distance is very close, and the application scenarios are very limited. There is an urgent need to find a new target location method and expand the application scenarios.

**Table 1.** Experimental results of GNSS-based passive radar. (LNA: low noise amplifier; MTI: moving target indicator; CAF: cross ambiguity function; SISAR: shadow inverse synthetic aperture radar).

| Number | Country | Object | Detection Method | Receive Gain (dB) | Accumulation Time | Detection Range |
|--------|---------|--------|------------------|-------------------|-------------------|-----------------|
| 1 | Germany [3] | Space Station | Signal disturbance Analysis | −5~−3 | 20 ms | 400 km |
| 2 | Britain [4] | Merchant ship (180 × 25 m) | Long integration time MTI processing | 15 | 50 s | 3 km |
| 3 | Britain [7] | Helicopter | Micro-Doppler signature Analysis | 25 (with LNA) | 21 ms | 2.4 km |
| 4 | Italy [8,9] | Merchant ship (150 × 23 m) | Long integration time; Track-before-detect | 15 | 10 s | 1.8 km |
| 5 | Spain [10] | Car | Coherent integration of consecutive CAFs | 35 (with LNA) | 1s | 60 m |
| 6 | China [11] | A321/A330 | SISAR Imaging | 4 | 20 ms | 1 km |

In bistatic radar, when the bistatic angle reaches 180 degrees, the radar cross section (RCS) of the target increases sharply [12–19]. This characteristic can be utilized in FSR. Regarding the parameter estimation in FSR, Professors Cherniakov [20] and Hu [21] have studied some algorithms. However, these algorithms require the arrival azimuth or pitch angle of the forward scattering echo. Therefore, the large aperture multi-beam antenna must be adopted, which causes a high cost-effectiveness ratio in the practical application.

If only the flight altitude of the aircraft target (usually less than 10 km) has to be determined, the problem of the low power limitation for GNSS-based passive radar can be reduced. Studies have shown that there was an apparent disturbance of echo when the target crossed the baselines [22–25]. By observing disturbances of echoes at multiple receivers, we can detect moving targets.

In this paper, the GNSS-based FSR architecture is proposed. Available target position estimation can be obtained by using the instant times that the target crosses the baselines. It is possible to find an effective method to estimate the crossing time by analyzing the characteristics of echo in the time domain, frequency domain, and time-frequency domain. In addition, the receivers (such as GPS receivers) have a simple structure when only measuring the time. A parameter estimation method based on crossing time was studied in [26], but the transmitters and receivers were required to be deployed at equal intervals and in parallel. The method in this paper removes this limitation. In addition, the significance of the crossing time is clarified, and the minimum configuration of the system is derived theoretically. Various error sources affecting the performance of parameter estimation are analyzed through Monte Carlo simulations, including the crossing time error, station location error, transmitting/receiving station deployment, and target height. According to the simulation results, the application scenario of the system is given, which provides a reference for the deployment of the actual FSR network.

The rest of this paper is organized as follows. Section 2 defines the uniformly accelerated motion observation model of the FSR network. Section 3 describes the parameter estimation method. Section 4 shows some simulation results with different errors which shows the potential application scenario of the system. Our conclusions are given in Section 5.

## 2. Observation Model in FSR Network

As shown in Figure 1, the FSR network consists of the radiation source of widely existing satellite navigation signals and sparse receiving arrays on the ground. FSR is a particular case of a multistatic radar configuration. It takes full advantage of the forward scattering area of the bistatic radar cross section for target detection, and has a strong anti-stealth ability. Moreover, the network has the following prominent advantages:

(1) Satellite navigation signals exist widely, and any point on Earth is illuminated typically by 20–30 satellites simultaneously from different angles;

(2) Satellite navigation signals have high time-frequency synchronization accuracy;

(3) There is no need to make a single passive antenna very large, so it is simple and easy to deploy;

(4) Only the height information needs to be detected, which alleviates the problem of long-distance detection.

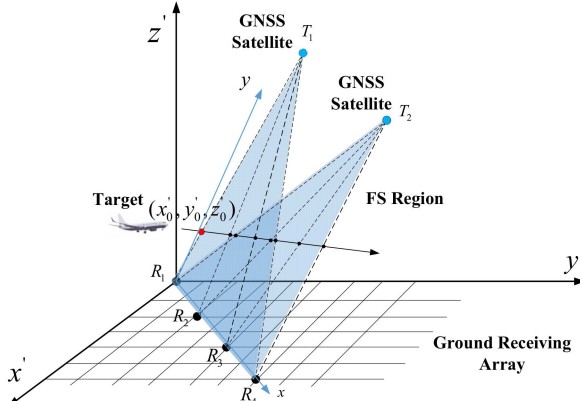

**Figure 1.** FSR network structure.

When the target crosses the connection between the transmitter station and the receiver, the ideal forward scattering state is formed. The existence of the target can be confirmed by the disturbance of the receiving signal, and the movement of the target can be perceived continuously by the detection of multiple receivers. As for the processing of the receiving signal, the conventional GNSS signal processing method could be used. We can extract the C/A code correlation value for every signal integration interval and confirm the satellite serial number. Furthermore, there is no range resolution, but rather Doppler resolution in the forward scattering region. The signal's Doppler resolution has been proven to be a linear function of time [25] and approaches zero when the target crosses the baseline, and this feature can be used to estimate the crossing time.

## 3. Proposed Parameter Estimation Method

The coordinates of the satellites and receiving stations are three-dimensional space coordinates in the real application scenario. Describing the target position in a three-dimensional space can reflect the practical value of the algorithm much better, as shown in Figure 1, assuming that the baselines that the target has crossed correspond to $N$ transmitters and $K$ receivers. According to the order in which the target crosses, the transmitters are numbered $T_n(x'_{tn}, y'_{tn}, z'_{tn})$, $(n = 1, 2, \ldots, N)$, and the receivers are numbered $R_k(x'_{rk}, y'_{rk}, z'_{rk})$, $(k = 1, 2, \ldots, K)$. The coordinate system is taken to be centered in $R_1$. The coordinate of the intersection of the target and the first baseline is $(x'_0, y'_0, z'_0)$, and $(x_0, y_0, z_0)$ used in Section 4. To reduce the impact of satellite motion, the observation time should be limited, and assume that the crossed baselines are approximately in the same plane. The error caused by this approximation can be converted into the crossing time error and the station location error.

Then, we convert the three-dimensional (3D) coordinate $(x\prime, y\prime, z\prime)$ to a two-dimensional (2D) coordinate $(x,y)$ as shown in Figure 2. Assuming the equation of the plane is $ax\prime + by\prime + cz\prime + d = 0$, the receiver $R_1$ is set as the origin $O$, with the receivers $R_1, R_2, \ldots, R_K$ placed along the $x$-axis, while the line passing through the origin and being perpendicular to the $x$-axis is taken as the $y$-axis. The direction vectors of the $x$-axis and $y$-axis are $\overrightarrow{x} = (m_x, n_x, p_x)$ and $\overrightarrow{y} = (m_y, n_y, p_y)$, respectively. Then, the 2D coordinate can be obtained by calculating the distance from each point to the $x$-axis and $y$-axis.

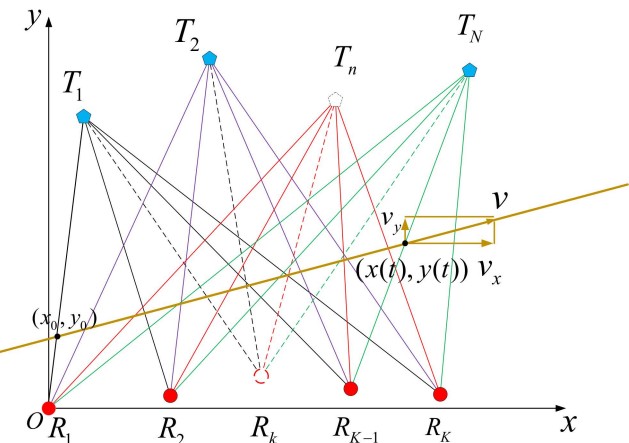

**Figure 2.** Positioning plane geometry structure.

Taking the solution of the ordinate of $T_n(x'_{tn}, y'_{tn}, z'_{tn})$ as an example, the orthogonal projection of $T_n$ on the $x$-axis is

$$\begin{cases} x'_d = m_x t \\ y'_d = n_x t \\ z'_d = p_x t \end{cases} \tag{1}$$

where $t = \frac{m_x x'_{tn} + n_x y'_{tn} + p_x z'_{tn}}{m_x^2 + n_x^2 + p_x^2}$.

Therefore, the vertical ordinate of $T_n(x'_{tn}, y'_{tn}, z'_{tn})$ in the 2D coordinate system is $y_{tn} = \sqrt{(x'_{tn} - x_d)^2 + (y'_{tn} - y_d)^2 + (z'_{tn} - z_d)^2}$.

For the consideration of simplification, the coordinate of the $n_{\text{th}}$ transmitter is $\mathbf{T}_n = [x_{tn}, y_{tn}]^{\mathrm{T}}$, $(n = 1, 2, \dots, N)$, and the coordinate of the $k$th receiver is $\mathbf{R}_k = [x_{rk}, y_{rk}]^{\mathrm{T}}$, $(k = 1, 2, \dots, K)$. Assuming that the target moves in a straight line with uniform acceleration $a$ in the $xOy$ plane, the initial velocity is $v$, $v_x$ is the velocity in the $x$-direction, $v_y$ is the velocity in the $y$-direction, and the initial position is $(x_0, y_0)$. The target coordinates change with time as follows:

$$x(t) = x_0 + v_x t + \frac{1}{2} a_x t^2 \,,\; y(t) = y_0 + v_y t + \frac{1}{2} a_y t^2 \tag{2}$$

The pair $(n,k)$ corresponds to the $n$th transmitter and the $k$th receiver. The start time $(t_{nk} = 0)$ is the moment when the target crosses the first baseline. The target crosses the individual baseline $(n,k)$ at time $t_{nk}$, and thus the following equation can be derived:

$$(y_{tn} - y_{rk})(x_0 + v_x t_{nk} + \frac{1}{2} a_x t_{nk}^2) - (y_{tn} - y_{rk})x_{tn} = (x_{rk} - x_{tn})y_{tn} - (x_{rk} - x_{tn})(y_0 + v_y t_{nk} + \frac{1}{2} a_y t_{nk}^2) \tag{3}$$

where $t_{nk}$ represents the time required for the target to move from the initial position to the current position. For the first baseline, $t_{nk} = 0$ and the initial position $(x_0, y_0)$ satisfies the following equation:

$$(y_{tn} - y_{rk})x_0 + (x_{rk} - x_{tn})y_0 - (y_{tn} - y_{rk})x_{tn} = x_{rk}y_{tn} - x_{tn}y_{rk} \tag{4}$$

In a matrix form, Equation (3) can be expressed as

$$\left[ y_{tn} - y_{rk} \;\; x_{rk} - x_{tn} \;\; (y_{tn} - y_{rk})t_{nk} \;\; (x_{rk} - x_{tn})t_{nk} \;\; \frac{1}{2}(y_{tn} - y_{rk})t_{nk}^2 \;\; \frac{1}{2}(x_{rk} - x_{tn})t_{nk}^2 \right] \begin{bmatrix} x_0 \\ y_0 \\ v_x \\ v_y \\ a_x \\ a_y \end{bmatrix} = x_{rk}y_{tn} - x_{tn}y_{rk} \tag{5}$$

where the six unknowns (target position, velocities, and accelerations) are arranged in a vector. Assuming that $P$ intersections are available, and $P \geq N_U$, with $N_U$ being the number of unknowns, the unknown vector will be the solution of a linear equation set, as shown in Equation (6)

$$
\begin{bmatrix}
y_{tn_1} - y_{rk_1} & x_{rk_1} - x_{tn_1} & 0 & 0 & 0 & 0 \\
y_{tn_2} - y_{rk_2} & x_{rk_2} - x_{tn_2} & (y_{tn_2} - y_{rk_2})t_{n_2k_2} & (x_{rk_2} - x_{tn_2})t_{n_2k_2} & \frac{1}{2}(y_{tn_2} - y_{rk_2})t_{n_2k_2}^2 & \frac{1}{2}(x_{rk_2} - x_{tn_2})t_{n_2k_2}^2 \\
y_{tn_3} - y_{rk_3} & x_{rk_3} - x_{tn_3} & (y_{tn_3} - y_{rk_3})t_{n_3k_3} & (x_{rk_3} - x_{tn_3})t_{n_3k_3} & \frac{1}{2}(y_{tn_3} - y_{rk_3})t_{n_3k_3}^2 & \frac{1}{2}(x_{rk_3} - x_{tn_3})t_{n_3k_3}^2 \\
\multicolumn{6}{c}{\cdots\cdots} \\
y_{tn_P} - y_{rk_P} & x_{rk_P} - x_{tn_P} & (y_{tn_P} - y_{rk_P})t_{n_pk_p} & (x_{rk_P} - x_{tn_P})t_{n_pk_p} & \frac{1}{2}(y_{tn_P} - y_{rk_P})t_{n_pk_p}^2 & \frac{1}{2}(x_{rk_P} - x_{tn_P})t_{n_pk_p}^2
\end{bmatrix}
\begin{bmatrix} x_0 \\ y_0 \\ v_x \\ v_y \\ a_x \\ a_y \end{bmatrix}
=
\begin{bmatrix}
x_{rk_1}y_{tn_1} - x_{tn_1}y_{rk_1} \\
x_{rk_2}y_{tn_2} - x_{tn_2}y_{rk_2} \\
x_{rk_3}y_{tn_3} - x_{tn_3}y_{rk_3} \\
\vdots \\
x_{rk_P}y_{tn_P} - x_{tn_P}y_{rk_P}
\end{bmatrix}
\tag{6}
$$

For the sake of simplicity, Equation (6) can be expressed in a matrix form as follows.

$$CX = F \tag{7}$$

where

$$
C =
\begin{bmatrix}
y_{tn_1} - y_{rk_1} & x_{rk_1} - x_{tn_1} & 0 & 0 & 0 & 0 \\
y_{tn_2} - y_{rk_2} & x_{rk_2} - x_{tn_2} & (y_{tn_2} - y_{rk_2})t_{n_2k_2} & (x_{rk_2} - x_{tn_2})t_{n_2k_2} & \frac{1}{2}(y_{tn_2} - y_{rk_2})t_{n_2k_2}^2 & \frac{1}{2}(x_{rk_2} - x_{tn_2})t_{n_2k_2}^2 \\
y_{tn_3} - y_{rk_3} & x_{rk_3} - x_{tn_3} & (y_{tn_3} - y_{rk_3})t_{n_3k_3} & (x_{rk_3} - x_{tn_3})t_{n_3k_3} & \frac{1}{2}(y_{tn_3} - y_{rk_3})t_{n_3k_3}^2 & \frac{1}{2}(x_{rk_3} - x_{tn_3})t_{n_3k_3}^2 \\
\multicolumn{6}{c}{\cdots\cdots} \\
y_{tn_P} - y_{rk_P} & x_{rk_P} - x_{tn_P} & (y_{tn_P} - y_{rk_P})t_{n_pk_p} & (x_{rk_P} - x_{tn_P})t_{n_pk_p} & \frac{1}{2}(y_{tn_P} - y_{rk_P})t_{n_pk_p}^2 & \frac{1}{2}(x_{rk_P} - x_{tn_P})t_{n_pk_p}^2
\end{bmatrix}
$$

$$
X =
\begin{bmatrix} x_0 \\ y_0 \\ v_x \\ v_y \\ a_x \\ a_y \end{bmatrix}, \text{ and } F =
\begin{bmatrix}
x_{rk_1}y_{tn_1} - x_{tn_1}y_{rk_1} \\
x_{rk_2}y_{tn_2} - x_{tn_2}y_{rk_2} \\
x_{rk_3}y_{tn_3} - x_{tn_3}y_{rk_3} \\
\vdots \\
x_{rk_P}y_{tn_P} - x_{tn_P}y_{rk_P}
\end{bmatrix}
$$

$F$ is constructed by the locations of the navigation satellites and the receivers, which can be obtained by the mature methods in the navigation field. Since the measurement accuracy of the navigation satellites and the receiver locations in $F$ is high, this equation can be solved by the least squares method. When $C^T C$ is full rank, the estimated value of target motion parameters is given by

$$X = (C^T C)^{-1} C^T F \tag{8}$$

Equation (8) can be solved by the square root method, the successive over-relaxation (SOR) method, the Newton iterative method, and so on. When the Newton iterative method is used, the initial iterative value can be obtained by randomly selecting enough equations.

According to the definition of matrix $C$, when the abscissa of the receiving station and the transmitting station are equal ($x_{rk_i} = y_{tn_i}$), the element in the second column of the $i$-th row of the matrix is zero. If that is satisfied for I = 1~P, it means that the system is composed of multiple baselines perpendicular to the horizontal plane and parallel to each other. At this time, parameter estimation cannot be carried out.

Therefore, the estimated result of the target initial position is given by

$$
\begin{cases}
x_0 = [(C^T C)^{-1} C^T F]_1 \\
y_0 = [(C^T C)^{-1} C^T F]_2
\end{cases}
\tag{9}
$$

For a "one-transmitter and six-receiver" system, the rank of $C$ is five (rank deficit of 1). Thus, it is not possible to use T-R$^N$ or T$^N$-R mode for parameter estimation in the FSR network. The minimum configuration of the system is "two-transmitter and three-receiver" or "three-transmitter and two-receiver". The details are given in the Appendix A.

Then, the initial position estimation result in 2D can be converted into 3D space as follows:

$$\begin{cases} x_0' = x_0 \cos \alpha_x + y_0 \cos \alpha_y \\ y_0' = x_0 \cos \beta_x + y_0 \cos \beta_y \\ z_0' = x_0 \cos \gamma_x + y_0 \cos \gamma_y \end{cases} \tag{10}$$

where $(\alpha_x, \beta_x, \gamma_x)$ and $(\alpha_y, \beta_y, \gamma_y)$ correspond to the direction angles of the *x*-axis and *y*-axis direction vectors in the 3D coordinate system, respectively.

From (7), it can be inferred that the factors that affect the estimation accuracy are composed of the crossing time error and station location error. Considering the technical performance of the actual system, the error sources and approximate error magnitude are given in Table 2 (assuming each error is independent of each other and obeys the zero-mean Gaussian distribution).

**Table 2.** Systematic errors.

| Error Source | Main Factor | Standard Deviation |
|---|---|---|
| Crossing time measurement error | Space noise environment, target crossing inaccurately, RCS fluctuation, system synchronization error, etc. | <1300 μs |
| Receiver station site error | Station position measurement error | 3~5 m |
| Transmitter station site error | Satellite motion, satellite position measurement error | <1000 m |

Where the position of the transmitting/receiving station can be obtained by theodolite or navigation satellite positioning in advance. The errors $e_x$ and $e_y$ measured by theodolite are about 5~8 m, and $e_h$ is about 1 m. The positioning error of GPS is generally about 3 m.

The measurement accuracy of crossing time is analyzed in 23 through incoherent processing. The extracted multiple crossing times are regarded as independent Gaussian random variables. The Cramer Rao lower bound is expressed as

$$\sigma_{n,k} = \frac{1}{\beta_{n,k} \sqrt{\text{SNR}_{n,k}}} \tag{11}$$

where $\beta_{n,k}$ is the effective bandwidth of the target signal on the baseline, and $\text{SNR}_{n,k}$ is the signal-to-noise ratio on the baseline $(n, k)$ after coherent integration within the processing interval. $n$ is the transmitting station number and $k$ is the receiving station number. Considering the FSR network based on Beidou, $\beta_{n,k}$ = 20.46 MHz, $\text{SNR}_{n,k}$ = 13 dB, so $\sigma_{n,k} \approx 13.6$ ns. Taking other factors in Table 2 into account, the error standard deviation is enlarged to 1300 μs.

## 4. Simulation Verification and Error Analysis

In this section, the impacts of the crossing time error and the location errors of the receiver and the transmitter on the parameter estimation accuracy are analyzed through Monte Carlo simulation, and the target positioning capability of the actual system is determined. The system simulation parameters are set as shown in Table 3.

**Table 3.** Simulation parameter configuration.

| | | |
|---|---|---|
| Target | Flight trajectory (2D) (m) | Y = 200x + 7000 |
| | Velocity (m/s) | 350 |
| | Acceleration (m/s$^2$) | 50 |
| Receiver | $R_1$ (m) | (0,0,0) |
| | $R_2$ (m) | (400,400,10) |
| | $R_3$ (m) | (800,800,−10) |
| Transmitter | $T_1$ (km) | (100,200,20,200) |
| | $T_2$ (km) | (200,300,20,200) |
| Errors | The standard deviation of crossing time error (us) | 300 |
| | The standard deviation of receiver station site error (m) | 5 |
| | The standard deviation of transmitter station site error (m) | 1000 |

Here, the transmitting/receiving error is a three-dimensional space error. For example, the standard deviation of satellite position error is set to 1000 m, and the random error with a standard deviation of 1000 m is added to the coordinates of the satellite in three directions. Moreover, it is assumed that each error follows a Gaussian distribution with a mean of zero. Monte Carlo simulation is used for statistical error analysis.

The target position estimate refers to the coordinates of the intersection of the target trajectory and the first baseline. In addition, the error of each dimension of the three-dimensional coordinates is defined as follows:

$$RMSE_{x_0} = \sqrt{\frac{1}{N}\sum_{i=1}^{N}(x_i - x_0)^2} \tag{12}$$

$$RMSE_{y_0} = \sqrt{\frac{1}{N}\sum_{i=1}^{N}(y_i - y_0)^2} \tag{13}$$

$$RMSE_{z_0} = \sqrt{\frac{1}{N}\sum_{i=1}^{N}(z_i - z_0)^2} \tag{14}$$

where $(x_0, y_0, z_0)$ is the 3D coordinate of the real initial baseline intersection, $(x_i, y_i, z_i)$ is the estimated value of three-dimensional coordinates obtained from the $i$-th Monte Carlo simulation, and $N$ is the number of Monte Carlo simulations.

A.    Influence of measurement error on estimation accuracy

Figure 3 shows the root mean square errors (RMSEs) of the target location, velocity, and acceleration with different crossing time errors, receiver and transmitter station site errors. It can be obtained that all the RMSEs increase with the crossing time error, receiver and transmitter station site errors. The RMSEs of the initial location are shown in Figure 3a,c,e. In accordance with the systematic error in Table 2, it can be seen that the error is less than 100 m when the crossing time error is less than 1.3 ms, the error is less than 280 m when the receiver station site error is less than 5 m, and the error is less than 100 m when the transmitter station site error is less than 1000 m. In the real scene, the current position measurement accuracy of the navigation satellites and receivers can achieve the meter level. Through some special processing, higher accuracy can be obtained. In Figure 3b,d,f, the RMSEs of the velocity and acceleration estimates exceed 200 m/s and

200 m/s$^2$, respectively, when the crossing time error reaches 1.3 ms and the receiver station site error reaches 5 m. Those estimates are larger than the true values.

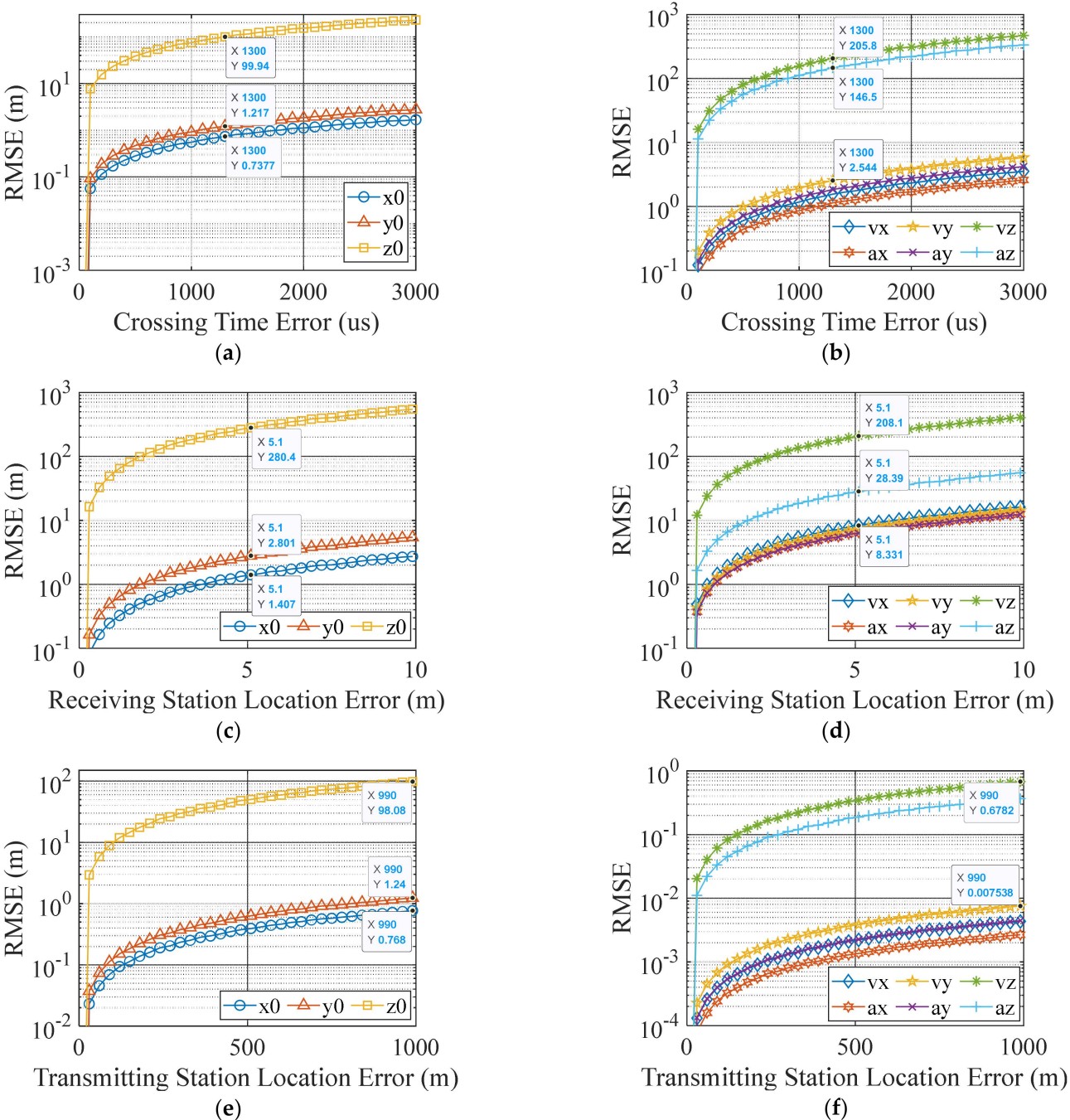

**Figure 3.** Influence of error sources on parameter estimation accuracy. (**a**) Initial position estimation. (**b**) Velocity and acceleration estimation. (**c**) Initial position estimation. (**d**) Velocity and acceleration estimation. (**e**) Initial position estimation. (**f**) Velocity and acceleration estimation.

In general, the RMSE of the target position estimate is in the order of 100 m. Additionally, the estimation error of the coordinate component in the *z*-direction is much larger than that in the *x*- and *y*-directions, which is the main component of the position error. Moreover, the measurement error greatly affects the estimation accuracy of velocity and acceleration. In the case of a large measurement error, the RMSE of acceleration estimate reaches an unacceptable level in the *z*-direction. In this condition, there is less reference significance to use the velocity and acceleration estimates in this model.

## B. Influence of system structure on estimation accuracy

The system sensitivity analysis consists of the different geometric structures of the FSR network and different heights. The following simulation shows the influence of the spacing between receivers, the spacing between transmitters, and target height on the positioning. The error setting is shown in Table 3, and the minimum system structure of $T^2$–$R^3$ is adopted. In the simulation, the variation range is 100–1000 m for receiver spacing, 10–1000 km for transmitter spacing, and 1000–10,000 m for target height. The results are shown in Figures 4 and 5. As shown in a, the RMSE of positioning decreases gradually with the receiver spacing. When the spacing exceeds 500 m, the error change is no longer obvious. Moreover, a larger receiver spacing is unfavorable for data transmission and synchronization. In Figure 4b, as the transmitter spacing increases, the RMSE of positioning decreases. In addition, the error reduction is no longer obvious when the spacing exceeds 70 km. To obtain a high-precision measurement of the crossing time and small RMSE of positioning, satellites far from each other with better signal quality should be selected as transmitters. In Figure 4c, the positioning error in the *z*-direction increases gradually with the target height, and we approximate the height as the *z* coordinate of the initial position of the target; however, the positioning RMSE remains within the acceptable range, and the error is less than 5% when converted into relative error,

$$RMSE_{z_0}/z_0 = \sqrt{\frac{1}{N}\sum_{i=1}^{N}(z_i - z_0)^2}/z_0 \approx 420.5/10,000 = 4.205\%,$$ where $N$ is the number of

Monte Carlo simulations, and *z* is the *z*-axis coordinate of the initial position of the target obtained by each simulation. Therefore, high-precision positioning can be achieved for general target heights.

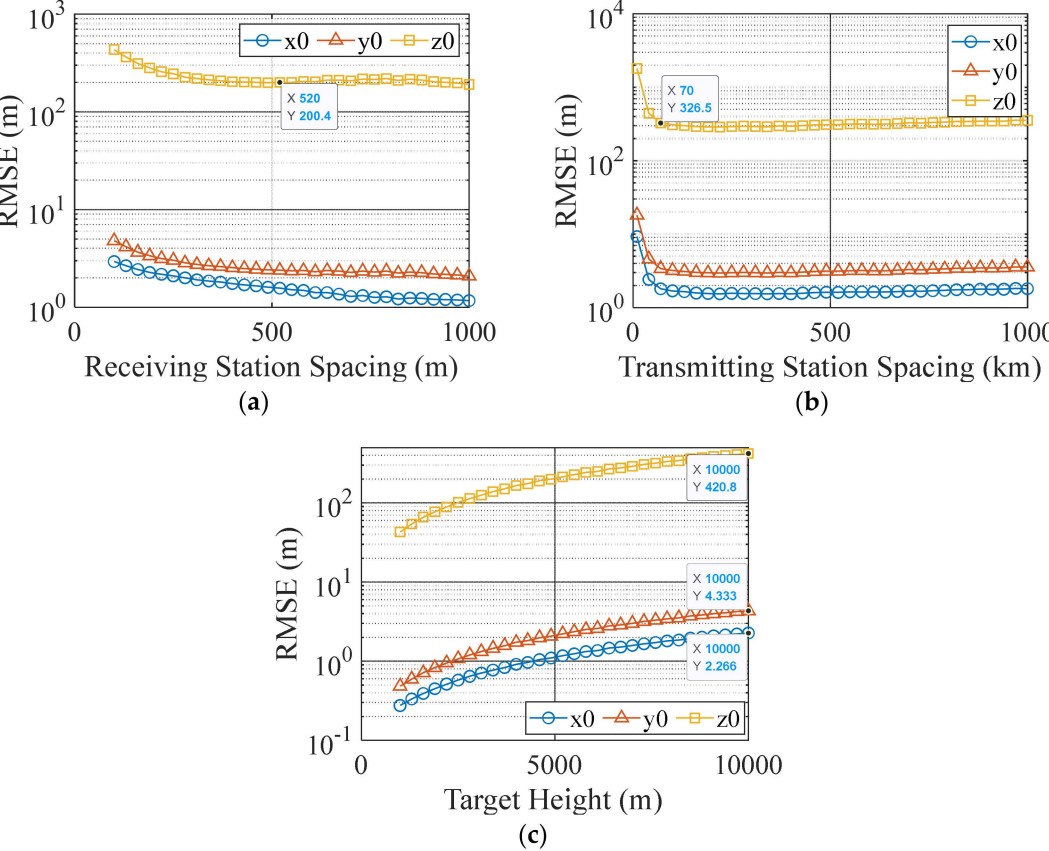

**Figure 4.** Influence of system structure on the accuracy of the initial position. (**a**) Influence of receiver spacing. (**b**) Influence of transmitter spacing. (**c**) Influence of target height.

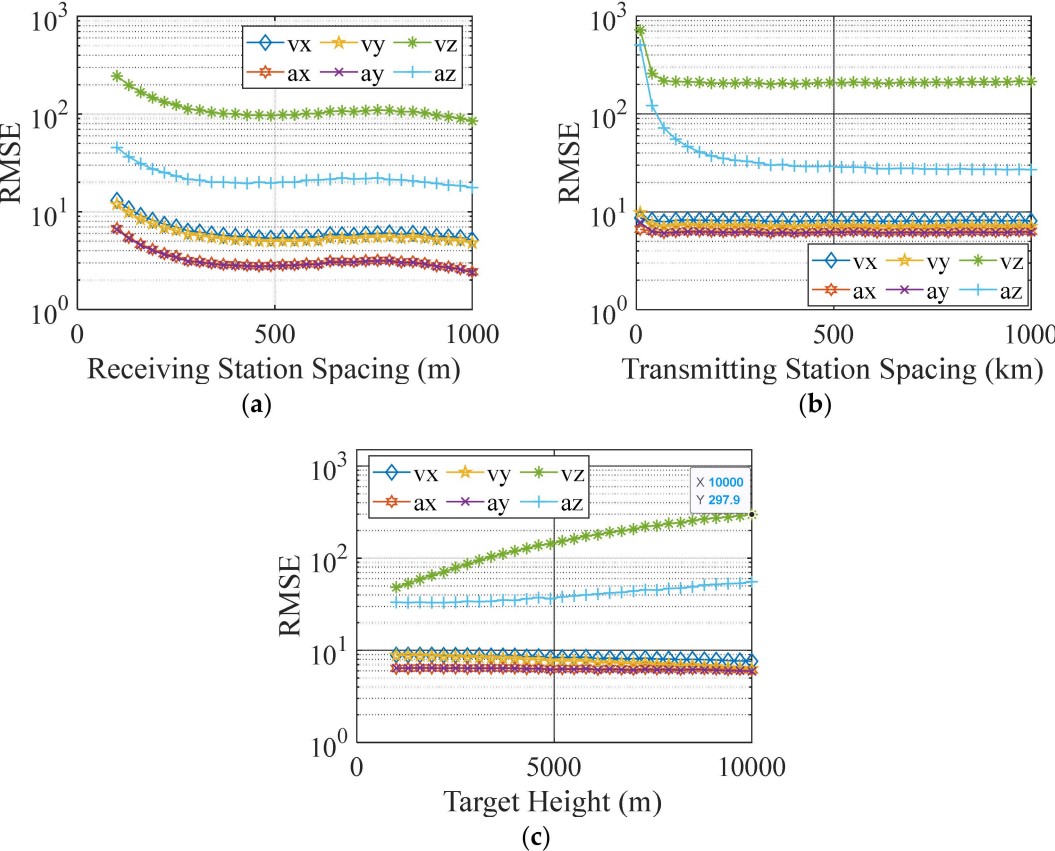

**Figure 5.** Influence of system structure on parameter estimation accuracy. (**a**) Influence of receiver spacing. (**b**) Influence of transmitter spacing. (**c**) Influence of target height.

Figure 5 shows the RMSEs of the target velocity and acceleration. As shown in Figure 5a,b, the RMSEs of velocity and acceleration estimates decline with the receiver spacing and transmitter spacing. The RMSE of acceleration estimate in each direction reduces to approximately 10 m/s$^2$, while the RMSE of velocity estimate in the *z*-direction exceeds 100 m/s. It is difficult to meet the practical requirements by changing the observation configuration. In Figure 5c, the RMSEs of velocity and acceleration estimates in the *x*-direction and ydirection hardly change with the increase in target height, while that in the *z*-direction increases gradually. The RMSE of the acceleration estimate is more than 100 m/s$^2$ in the *z*-direction, so the estimated results cannot be utilized for the target track.

C.    Influence of target motion on estimation accuracy

The following simulation shows the influence of the target initial velocity and acceleration on the positioning. The error setting is shown in Table 3, and the minimum system structure of T$^2$–R$^3$ is adopted. In the simulation, the variation range is 100–1000 m/s for initial velocity and 10–100 m/s$^2$ for acceleration. Figure 6a shows the RMSE of the coordinate component estimate in each direction with different initial velocities. The RMSE of positioning decreases gradually with the initial velocity increasing. When the initial velocity exceeds 500 m/s, the error change is no longer obvious. Additionally, the RMSE in the *z*-direction is basically around 260 m. Figure 6b shows the RMSE of positioning with different accelerations. The initial velocity is set to 350 m/s. As we can see, the RMSE of positioning gradually increases with increasing acceleration; however, the increase is limited and the impact can be ignored. Therefore, the target acceleration has little effect on the initial position estimation accuracy.

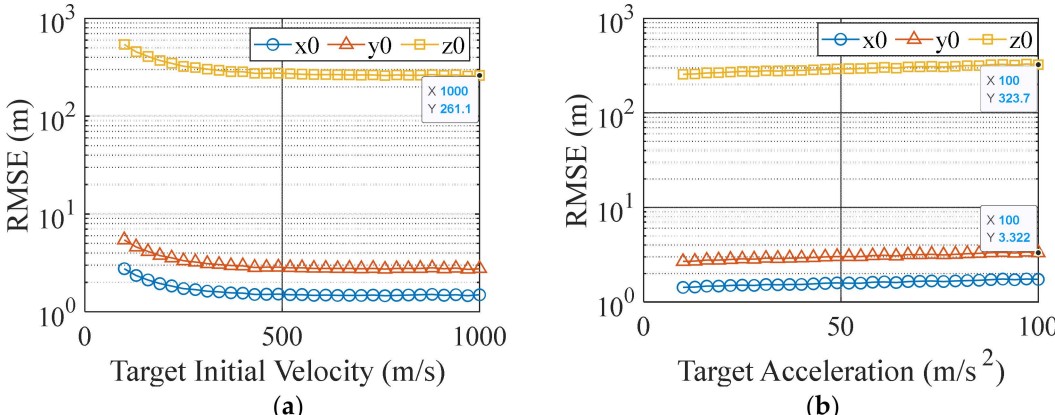

**Figure 6.** Influence of target motion on parameter estimation accuracy. (**a**) Influence of initial velocity ($a$ = 50 m/s$^2$). (**b**) Influence of acceleration (v = 350 m/s).

Summarizing the above simulation analysis, we can obtain the following conclusions:

(1) The location of a moving target with uniform acceleration can be estimated only using the multiple crossing times in a FSR network. Under the condition of existing systematic error, the RMSE of positioning is less than 100 m.

(2) The system structure has an impact on target positioning and motion parameter estimation. It is better to set the receiver spacing to about 500 m and choose the transmitters with good signal quality and large spacings.

(3) Target velocity and acceleration have an impact on target positioning and motion parameter estimation. A high velocity leads to a shorter time for the target to continuously cross multiple baselines and a higher accuracy of target position estimation, while the impact of acceleration can be ignored.

(4) The velocity and acceleration estimate can be referenced when the systematic error is small. However, it is necessary to use position tracking filtering to obtain higher accuracy when the error is large.

## 5. Conclusions

In this paper, a motion parameter estimation method only using the crossing times is proposed in the FSR network with GNSS satellite as the radiation source. Additionally, the minimum network configuration is derived theoretically. The effect of error sources and system structure on the parameter estimation accuracy is analyzed. The simulation results indicate that the initial position estimation error is in the order of 100 m for moving target with uniform acceleration under the condition of actual measurement error. This magnitude of error meets the accuracy requirements of early warning detection. The proposed method provides a potential application scenario for passive radar based on GNSS. However, there is a problem that the error values of velocity and acceleration estimates are large. We could obtain higher accuracy by continuously positioning and tracking filtering. In future research, we will further quantitatively analyze the impact of the inaccurate crossing (the target trajectory has a certain distance from the baseline) on the crossing time measurement accuracy. Based on that, the acceptable inaccurate crossing could be determined, which can provide theoretical support for the optimal station layout. A Multi-baseline FSR experimental system is currently being designed and it will be used for the experimental verification of the proposed method later.

**Author Contributions:** Conceptualization, X.A. and Y.Z.; methodology, X.A. and Y.Z.; validation, Y.Z. writing—original draft preparation, X.A. and Y.Z.; writing—review and editing, Z.X. and F.Z.; supervision, F.Z. All authors have read and agreed to the published version of the manuscript.

**Funding:** This research was funded by [National Natural Science Foundation of China] grant number [62071475].

**Data Availability Statement:** Data underlying the results presented in this paper are not publicly available at this time but may be obtained from the authors upon reasonable request.

**Conflicts of Interest:** The authors declare no conflict of interest.

## Appendix A

Assuming that a FSR network is composed of only one transmitter and multiple receivers, multiple equations for the motion parameters of the uniformly accelerated motion target can be obtained. Since there are six unknown parameters, at least six equations are required. The simplest system is the "one-transmitter and six-receiver" system, as shown in Figure A1.

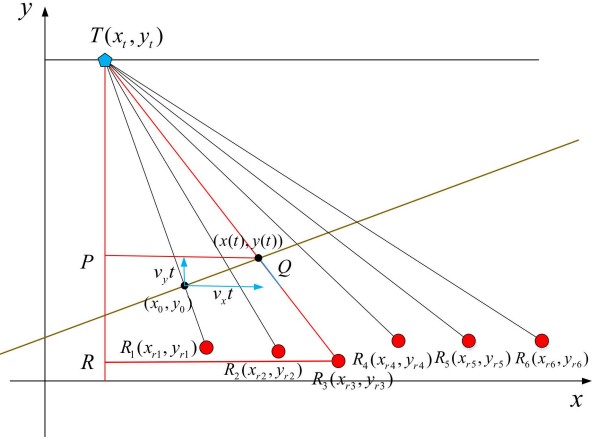

**Figure A1.** FSR system structure diagram of one-transmitter and six-receiver.

According to the target motion parameter equation derived in this paper, the positioning matrix $C$ is

$$
C = \begin{bmatrix}
y_t - y_{r1} & x_{r1} - x_t & 0 & 0 & 0 & 0 \\
y_t - y_{r2} & x_{r2} - x_t & (y_t - y_{r2})t_{12} & (x_{r2} - x_t)t_{12} & 0.5(y_t - y_{r2})t_{12}^2 & 0.5(x_{r2} - y_t)t_{12}^2 \\
y_t - y_{r3} & x_{r3} - x_t & (y_t - y_{r3})t_{13} & (x_{r3} - x_t)t_{13} & 0.5(y_t - y_{r3})t_{13}^2 & 0.5(x_{r3} - y_t)t_{13}^2 \\
y_t - y_{r4} & x_{r4} - x_t & (y_t - y_{r4})t_{14} & (x_{r4} - x_t)t_{14} & 0.5(y_t - y_{r4})t_{14}^2 & 0.5(x_{r4} - y_t)t_{14}^2 \\
y_t - y_{r5} & x_{r5} - x_t & (y_t - y_{r5})t_{15} & (x_{r5} - x_t)t_{15} & 0.5(y_t - y_{r5})t_{15}^2 & 0.5(x_{r5} - y_t)t_{15}^2 \\
y_t - y_{r6} & x_{r6} - x_t & (y_t - y_{r6})t_{16} & (x_{r6} - x_t)t_{16} & 0.5(y_t - y_{r6})t_{16}^2 & 0.5(x_{r6} - y_t)t_{16}^2
\end{bmatrix} \quad (A1)
$$

The first four columns of the matrix can be transformed as follows

$$
\begin{bmatrix}
y_t - y_{r1} & x_{r1} - x_t & 0 & 0 \\
y_t - y_{r2} & x_{r2} - x_t & (y_t - y_{r2})t_{12} & (x_{r2} - x_t)t_{12} \\
y_t - y_{r3} & x_{r3} - x_t & (y_t - y_{r3})t_{13} & (x_{r3} - x_t)t_{13} \\
y_t - y_{r4} & x_{r4} - x_t & (y_t - y_{r4})t_{14} & (x_{r4} - x_t)t_{14} \\
y_t - y_{r5} & x_{r5} - x_t & (y_t - y_{r5})t_{15} & (x_{r5} - x_t)t_{15} \\
y_t - y_{r6} & x_{r6} - x_t & (y_t - y_{r6})t_{16} & (x_{r6} - x_t)t_{16}
\end{bmatrix}
\xrightarrow{c_2 \times (y_t - y_0) - c_4 \times v_y}
$$

$$
\begin{bmatrix}
y_t - y_{r1} & (x_{r1} - x_t)(y_t - y_0) & 0 & 0 \\
y_t - y_{r2} & (x_{r2} - x_t)(y_t - y_0 - v_y t_{12}) & (y_t - y_{r2})t_{12} & (x_{r2} - x_t)t_{12} \\
y_t - y_{r3} & (x_{r3} - x_t)(y_t - y_0 - v_y t_{13}) & (y_t - y_{r3})t_{13} & (x_{r3} - x_t)t_{13} \\
y_t - y_{r4} & (x_{r4} - x_t)(y_t - y_0 - v_y t_{14}) & (y_t - y_{r4})t_{14} & (x_{r4} - x_t)t_{14} \\
y_t - y_{r5} & (x_{r5} - x_t)(y_t - y_0 - v_y t_{15}) & (y_t - y_{r5})t_{15} & (x_{r5} - x_t)t_{15} \\
y_t - y_{r6} & (x_{r6} - x_t)(y_t - y_0 - v_y t_{16}) & (y_t - y_{r6})t_{16} & (x_{r6} - x_t)t_{16}
\end{bmatrix}
\xrightarrow{c_2 - c_3 \times v_x}
$$

$$
\begin{bmatrix}
y_t - y_{r1} & (x_{r1} - x_t)(y_t - y_0) & 0 & 0 \\
y_t - y_{r2} & (x_{r2} - x_t)(y_t - y_0 - v_y t_{12}) - (y_t - y_{r2})v_x t_{12} & (y_t - y_{r2})t_{12} & (x_{r2} - x_t)t_{12} \\
y_t - y_{r3} & (x_{r3} - x_t)(y_t - y_0 - v_y t_{13}) - (y_t - y_{r3})v_x t_{13} & (y_t - y_{r3})t_{13} & (x_{r3} - x_t)t_{13} \\
y_t - y_{r4} & (x_{r4} - x_t)(y_t - y_0 - v_y t_{14}) - (y_t - y_{r4})v_x t_{14} & (y_t - y_{r4})t_{14} & (x_{r4} - x_t)t_{14} \\
y_t - y_{r5} & (x_{r5} - x_t)(y_t - y_0 - v_y t_{15}) - (y_t - y_{r5})v_x t_{15} & (y_t - y_{r5})t_{13} & (x_{r5} - x_t)t_{13} \\
y_t - y_{r6} & (x_{r6} - x_t)(y_t - y_0 - v_y t_{16}) - (y_t - y_{r6})v_x t_{16} & (y_t - y_{r6})t_{14} & (x_{r6} - x_t)t_{14}
\end{bmatrix}
\xrightarrow{c_2 - c_1 \times (x_0 - x_t)} \quad (A2)
$$

$$
\begin{bmatrix}
y_t - y_{r1} & (x_{r1} - x_t)(y_t - y_0) - (y_t - y_{r1})(x_0 - x_t) & 0 & 0 \\
y_t - y_{r2} & (x_{r2} - x_t)(y_t - y_0 - v_y t_{12}) - (y_t - y_{r2})(x_0 - x_t + v_x t_{12}) & (y_t - y_{r2})t_{12} & (x_{r2} - x_t)t_{12} \\
y_t - y_{r3} & (x_{r3} - x_t)(y_t - y_0 - v_y t_{13}) - (y_t - y_{r3})(x_0 - x_t + v_x t_{13}) & (y_t - y_{r3})t_{13} & (x_{r3} - x_t)t_{13} \\
y_t - y_{r4} & (x_{r4} - x_t)(y_t - y_0 - v_y t_{14}) - (y_t - y_{r4})(x_0 - x_t + v_x t_{14}) & (y_t - y_{r4})t_{14} & (x_{r4} - x_t)t_{14} \\
y_t - y_{r5} & (x_{r5} - x_t)(y_t - y_0 - v_y t_{15}) - (y_t - y_{r5})(x_0 - x_t + v_x t_{15}) & (y_t - y_{r5})t_{15} & (x_{r5} - x_t)t_{15} \\
y_t - y_{r6} & (x_{r6} - x_t)(y_t - y_0 - v_y t_{16}) - (y_t - y_{r6})(x_0 - x_t + v_x t_{16}) & (y_t - y_{r6})t_{16} & (x_{r6} - x_t)t_{16}
\end{bmatrix}
=
$$

$$
\begin{bmatrix}
y_t - y_{r1} & 0 & 0 & 0 \\
y_t - y_{r2} & 0 & (y_t - y_{r2})t_{12} & (x_{r2} - x_t)t_{12} \\
y_t - y_{r3} & 0 & (y_t - y_{r3})t_{13} & (x_{r3} - x_t)t_{13} \\
y_t - y_{r4} & 0 & (y_t - y_{r4})t_{14} & (x_{r4} - x_t)t_{14} \\
y_t - y_{r5} & 0 & (y_t - y_{r5})t_{15} & (x_{r5} - x_t)t_{15} \\
y_t - y_{r6} & 0 & (y_t - y_{r6})t_{16} & (x_{r6} - x_t)t_{16}
\end{bmatrix}
$$

The rank of the positioning matrix is 5, which is less than the number of unknown parameters. Moreover, for the number of receiver stations N > 6, the rank of the positioning matrix is also 5, so neither T-R$^N$ nor T$^N$-R mode can be used for motion parameter estimation. The minimum configuration of the FSR network can be determined as a "two-transmitter and three-receiver" or a "three-transmitter and two-receiver" structure.

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
