# Peer review of "Parameter Estimation for Uniformly Accelerating Moving Target in the Forward Scatter Radar Network"

_remotesensing, doi:10.3390/rs14041006_

Round 1

Reviewer 1 Report

The main drawback of this paper is a lack of validation of the proposed method using a real GNSS recorded signal.

Especially, from the title of the paper: "Parameter Estimation for Uniformly Accelerating Moving Target in the Forward Scatter Radar Network", I would suspect to see some real measurement results, not only simulations of the signal which character is based only on the geometry. And what about the low SNR level validation, which is typical in real GNSS signals? 

The research work has merit. My main doubt of this paper is lack of the algorithm validation using real measurement data. The article is mostly theoretical and lies on simulation results verification. When I read just the title of the paper, which is "Parameter Estimation for Uniformly Accelerating Moving Target in the Forward Scatter Radar Network" I was suspected to see, additionally to the simulation, also real results in the paper. This is a good practice in most radar-based papers that authors try to confirm their method works properly using measured signals. There often happens; then, in theory, everything works well, but theory fails when you have a real scenario. This is my main criticism for this paper.

Reviewer 2 Report

In this paper, a Parameter Estimation geometric model for a Uniformly Accelerating Moving Target in the Forward Scatter Radar Network is presented.
In order to verify the model, the results of the simulation are presented.

I can find many similar articles through a web search. The contribution point of this paper should be clearly indicated in the paper (at abstract or the end of introduction section).

Can this paper be subdued as an article type? Review the quantity.

It is unfamiliar to present experimental results from section 1. Of course, these are about existing methods, but using the word "experimental results" in the table caption can be misleading.

Analysis of existing studies is weak. Rather than presenting the experimental results of the existing method, it is necessary to analyze the summary of the existing method, Pros&Cons.

In the manuscript PDF file, the citaion for the figure or table is displayed as "Error! Reference source not found.", so it is difficult to understand the exact contents of the paper.

The parameter estimation strategy through matrix operation has been solidly developed.

Is the 4.205% calculated by dividing the RMSE by the height meaningful? Review presenting angular error with arctan.

What is the difference between the simulation result and the real result? It is recommended to present variables that may occur in real situations.

Reviewer 3 Report

see attached file.

Round 2

Reviewer 2 Report

All of the 1st round review comments were appropriately addressed.

Author Response

Thank you for your comments. We appreciate your efforts in reviewing our manuscript during this unprecedented and challenging time. We wish good health to you, your family, and your community.

Reviewer 3 Report

Dear authors

thanks for your reply and the revision with extensive answers.

However, I still see the equation 8) which is a least Square Solution, indeed. This is not the problem, however the 'Least square ' is minimising the squares of residuals of F (right side of the equation). But, to get a correct solution (in the sense of measurement theory)  one should minimize the residuals of the measurements also. (more generally speaking all values with uncertainties have to be considered as 'measurements' and their residuals square sum has to be minimized. In your used formalism this is not the case. 

Thus , to me there is some rigourousness missing.
